# Dynamical Latent Flow Matching for High-Performance Few-Shot Neural Adaptation

## Abstract

The primary objective of brain–computer interfaces (BCIs) is to establish a direct connection between neural activity and external devices. However, variability in neural recordings poses significant challenges to maintaining stable neural decoding with minimal recalibration. Existing neural decoding frameworks often fail to enable efficient few-shot adaptation, typically due to the constraints imposed by prior assumptions on latent variables or issues with training instability. Motivated by the flexibility and tractability of diffusion models, we propose the novel Dynamical Latent Flow Matching (DLFM) framework for high-performance few-shot neural adaptation. Our DLFM performs flow matching in dynamical latent spaces, leveraging preserved neural dynamics within the neural manifold. The probabilistic flexibility of DLFM effectively captures intrinsic features of dynamical patterns across heterogeneous sessions, significantly enhancing few-shot neural adaptation. The efficiency of DLFM for few-shot adaptation is validated on the Falcon benchmark, achieving competitive performance with only 60% calibration trials. Further experimental evaluation on the Neural Latents Benchmark 2021 demonstrates that DLFM ranks among the top two for forward prediction tasks across all web submissions. Additional interpretability analysis on the Lorenz attractor model and the Falcon dataset confirms that DLFM precisely identifies the intrinsic features of neural dynamics, thus facilitating efficient few-shot neural adaptation for neural decoding. Our DLFM framework emerges as a promising candidate for superior few-shot neural adaptation, advancing the practicality of real-world BCI systems.

## 1 Introduction

The goal of brain-computer interfaces (BCIs) is to establish a direct connection between the brain and external devices, offering promising avenues for neural rehabilitation in individuals with paralysis (Willett et al., 2021; Metzger et al., 2023; Willett et al., 2023). Nevertheless, variability in neural recordings, arising from physiological changes (Athalye et al., 2017) or device degradation (Woeppel et al., 2021), poses significant challenges to maintaining reliable decoding over time. Consequently, stable and high-performance neural adaptation is critical for maintaining long-term decoding with minimal recalibration (Karpowicz et al., 2024) in real-world BCI deployments. Self-supervised learning (Schneider et al., 2023) with Variational Autoencoders (VAEs) (Azabou et al., 2021; Schimel et al., 2022; Liu et al., 2021) and transformer-based neural foundation models (Ye & Pandarinath, 2021; Zhang et al., 2024; 2025) have been shown to extract latent variables from heterogeneous neural recordings, enabling practical BCI adaptation (Rafiei et al., 2022).

However, the performance of VAE- and transformer-based frameworks often deteriorates under few-shot neural adaptation with minimal recalibration. For example, the VAE-based Latent Factor Analysis via Dynamical Systems (LFADS) (Pandarinath et al., 2018) exhibits a pronounced performance decline under limited fine-tuning, with regression scores dropping below zero for five trials, as illustrated in Fig. 1(a). This may be attributed to the prior assumptions imposed on latent variables (Liu et al., 2025), which result in negative transfer shown in Fig. 1(b). Furthermore, the transformer-based foundation model MtM (Zhang et al., 2024) shows a reduced bits-per-second (bps) with few shots for high-dimensional prediction tasks, as depicted in Fig. 1(c). This issue stems from the training instability, as shown in Fig. 1(d), owing to both its large parameterization (Lai et al., 2022; Yang et al., 2024) and deterministic forecasters (Liu et al., 2025).

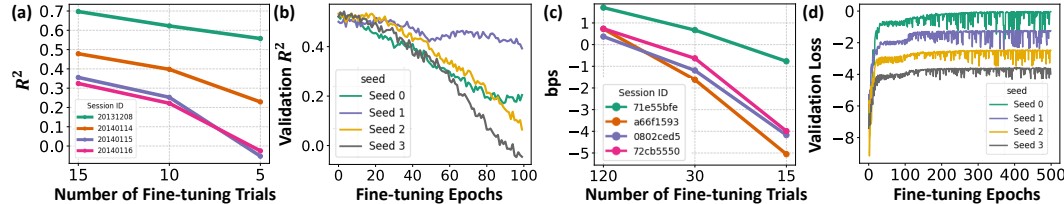

Figure 1: (a) The performance of LFADS on behavioral decoding across sessions with decreased fine-tuning trials. (b) Representative validation $R^2$ curves of LFADS across four different seeds on target sessions using 5 trials. (c) The bits-per-second (bps) performance of MtM on intra-region prediction across sessions varying with fine-tuning trials. (d) Representative validation loss curves of MtM using 30 fine-tuning trials.

This necessitates the development of a novel adaptation framework capable of learning flexible latent variable distributions through probabilistic generation. Motivated by the flexibility and tractability of diffusion models (Sohl-Dickstein et al., 2015), we propose performing flow matching in the dynamical latent space to capture stable dynamical patterns, thereby facilitating efficient few-shot adaptation. This flow matching framework is applicable in the neural dynamical latent space because preserved neural dynamical patterns persist across different individuals or sessions on low-dimensional manifolds (Safaie et al., 2023; Abbaspourazad et al., 2024).

Building on this concept, we introduce a novel neural decoding framework, Dynamical Latent Flow Matching (DLFM), which performs flow matching in the dynamical latent space by leveraging stable dynamical patterns within the neural manifold. The probabilistic flexibility of flow matching (Sohl-Dickstein et al., 2015) enables the learning of arbitrary distributions of dynamical patterns without prior assumptions, thereby enhancing transferability across heterogeneous sessions. Moreover, the tractable likelihood of flow matching allows parameter-efficient fine-tuning, contributing to the stable alignment under few-shot conditions. The efficacy of few-shot neural adaptation with DLFM is validated on the Falcon benchmark (Karpowicz et al., 2024), achieving competitive performance using only 60% of original calibration trials. The neural decoding performance of our DLFM framework is further comprehensively evaluated on the Neural Latents Benchmark 2021 (NLB21) (Pei et al., 2021), achieving a top-two ranking in the forward prediction tasks among all web submissions (Pei et al., 2021). Additional interpretability analysis on the Lorenz attractor model and the Falcon dataset confirms that DLFM precisely identifies the intrinsic features of neural dynamics, thus facilitating efficient few-shot neural adaptation for high-dimensional decoding.

Our DLFM stands as a promising framework for high-performance few-shot neural adaptation, advancing the practicality of real-world BCI systems. The main contributions of this paper are summarized as follows:

- **Dynamical Latent Flow Matching**: We propose a novel neural decoding framework based on flow matching in the dynamical latent space, which effectively captures stable features of dynamical patterns across heterogeneous sessions.
- **Efficient Few-shot Neural Adaptation**: The probabilistic flexibility and tractable likelihood of DLFM enable efficient alignment of dynamical latent patterns, attaining top-two ranking in forward bps on NLB21 and maintaining competitive performance on Falcon with only 60% of original calibration trials.
- **Experimental Validation**: We conducted extensive evaluations of DLFM on both synthetic and real neural datasets, demonstrating its superior performance in few-shot neural adaptation for high-dimensional neural decoding.

## 2 RELATED WORK

### 2.1 NEURAL ADAPTATION THROUGH SELF-SUPERVISED PRE-TRAINING

Neural adaptation is commonly achieved through self-supervised pre-training on heterogeneous neural recordings. VAE-based frameworks (Liu et al., 2021; Schimel et al., 2022) learn Gaussian latent variables, realizing tractable probabilistic computation. For example, LFADS Pandarinath et al. (2018); Karpowicz et al. (2025) employs sequential VAEs to obtain low-dimensional latent dynam-

ics from high-dimensional spike trains. In addition, neural foundation models have been proposed to extract shared latent variables from heterogeneous neural recordings (Jaegle et al., 2022; Azabou et al., 2023; 2025; Ryoo et al., 2025; Le et al., 2025) for neural adaptation. Several recent works are built on the neural data transformer (NDT) (Ye & Pandarinath, 2021) and its variants (Le & Shlizerman, 2022; Ye et al., 2023; 2025). MtM (Zhang et al., 2024; 2025) explores diverse masking schemes for multi-modal neural datasets. To address their decoding degradation of the above approaches under few-shot conditions, we propose a novel dynamical flow matching framework (DLFM) for efficient few-shot adaptation.

## 2.2 FLOW MATCHING

Flow matching (Lipman et al., 2022; Liu et al., 2022; Peebles & Xie, 2023; Geng et al., 2025) extends diffusion models by directly learning a velocity field that guides the transformation from noise to data. It achieves better efficiency for generation than that of denoising diffusion models (Tong et al., 2023). Conditional flow matching (Liu et al., 2023) further incorporates conditional features to control the generation process and has been applied in domains such as cell dynamics simulation (Atanackovic et al., 2025) and foundation models for time series (Liu et al., 2025). In the context of neural recordings, such models have been employed for data augmentation (Kapoor et al., 2024) by generating realistic synthetic neural activities. Furthermore, diffusion models have also been applied to align the latent dynamics between sessions and subjects (Wang et al., 2023). In this work, we employ conditional flow matching for self-supervised pre-training and few-shot neural adaptation, where its potential application in few-shot decoding remains underexplored.

## 3 METHODOLOGY

### 3.1 PROBLEM FORMULATION

We define the spiking signal dataset for self-supervised pretraining and few-shot neural adaptation as $\mathcal{D} = \{(x_1, y_1), \ldots, (x_n, y_n)\}$, where each $x_i(t)$ (for $t = 1, 2, \ldots, m_i$) represents a raw signal segment of length $m_i$. The segments are recorded from one or multiple sessions, typically divided by trials or short-time context windows. Each $x_i(t) \in \mathbb{R}^l$, where $l$ denotes the number of recording channels. The behavioral variables such as cursor velocities associated with $x_i(t)$ are denoted as $y_i(t) \in \mathbb{R}^d$, where $d$ represents dimensionality. For convenience, we omit the subscript $i$ and denote the segment and its behavioral variable by $x(t)$ and $y(t)$, respectively. Based on the dataset $\mathcal{D}$, we define pre-training datasets as $\mathcal{D}_p = \{(x_1^p, y_1^p), \ldots, (x_{n_p}^p, y_{n_p}^p)\}$, and fine-tuning datasets as $\mathcal{D}_f = \{(x_1^f, y_1^f), \ldots, (x_{n_f}^f, y_{n_f}^f)\}$, respectively. Similarly, we omit the subscript $i$ and denote the segments and their corresponding behavioral variables as $x^p$ and $y^p$ for the pre-training dataset, and $x^f$ and $y^f$ for the fine-tuning dataset, respectively.

### 3.2 OVERALL FRAMEWORK

In this work, we propose a novel DLFM framework that first incorporates conditional flow matching in the dynamical latent space for efficient few-shot neural adaptation. The proposed DLFM aims to capture features of dynamical patterns guided by conditional variables from spike trains. We introduce a coarse-to-fine refinement of noisy latent variables through flow-based generation, aiming to recover the underlying dynamical latent structures. The resulting dynamical latent variables are subsequently used to reconstruct raw spike trains and to perform downstream decoding tasks. Our DLFM exhibits superior performance in few-shot neural adaptation for decoding high-dimensional neural activity.

Building upon the aforementioned architecture, the DLFM framework comprises the two phases: self-supervised pre-training and parameter-efficient fine-tuning, as illustrated in Fig. 2. During the pre-training phase, the target dynamical latent space for flow matching is first constructed using an autoencoder. Conditioned on the features extracted from masked neural signals, dynamical latent flow matching performs a flow-based coarse-to-fine conditional reconstruction, mapping noisy latent variables to the target dynamical latent space. During the fine-tuning phase, the parameter-efficient fine-tuning is applied to the conditional network, achieving stable alignment of dynamical latent variables. The variables obtained have been validated to effectively capture the stable features of

dynamical patterns underlying neural populations, as verified by the Lorenz attractor in the subsequent sections. The overall framework of DLFM is depicted in Fig. 2 and will be detailed in the following. Further detailed architectures are provided in Section A.1.

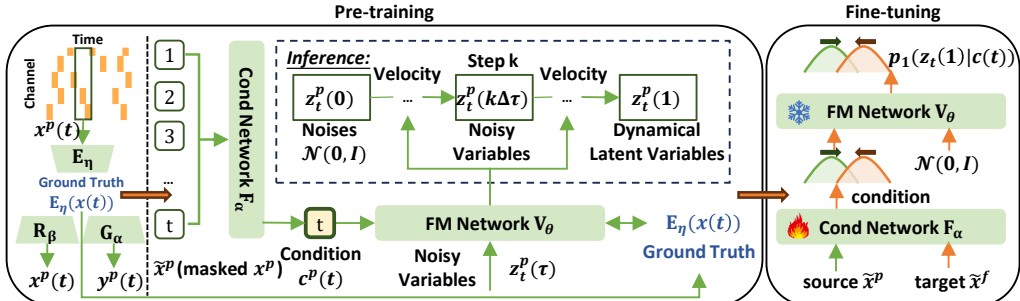

Figure 2: Workflow of the Dynamical Latent Flow Matching (DLFM) for few-shot neural adaptation. DLFM operates in two phases: (1) pre-training, where dynamical latent variables are captured via conditional flow-based generation, and (2) fine-tuning, where the conditional distribution of target latent variables is efficiently aligned.

### 3.2.1 DLFM-BASED SELF-SUPERVISED PRE-TRAINING

**Target Dynamical Latent Space** Flow matching (Lipman et al., 2022) typically adopts the $\ell_2$ norm to achieve consistency between generated and target variables. However, due to the inherently stochastic nature of neural recordings, directly applying deterministic loss functions in the original high-dimensional space often fails to capture this variability (Liu et al., 2025). To address this, we propose optimizing the target dynamical latent space of flow matching using the Poisson negative log-likelihood (PNLL) loss (Kapoor et al., 2024). This target dynamical latent space enables efficient reconstruction of Poisson-like spike trains from flow-based latent variables trained with deterministic loss functions. The target latent space is trained before flow matching and remains fixed during the following training.

Specifically, we construct the target dynamical latent space (Rombach et al., 2022) using an autoencoder with the encoder $E_\eta$ (parameterized by $\eta$) and the decoder $R_\beta$ (parameterized by $\beta$). The autoencoder is trained by minimizing the following PNLL loss over raw signal samples $x^p$ from the pre-training dataset:

$$\sum_t \left\{ R_\beta \left( E_\eta(x^p(t)) \right) - x^p(t) \odot \log \left[ R_\beta \left( E_\eta(x^p(t)) \right) \right] \right\}, \tag{1}$$

where $\odot$ indicates element-wise multiplication. $E_\eta(x^p(t))$ is the target dynamical latent variable, corresponding to $x^p(t)$. With the behavioral variable $y^p(t)$ and $E_\eta(x^p(t))$, a behavioral decoder $G_\gamma$ (with parameters $\gamma$) is trained in a supervised manner.

**Conditional Feature Extraction** After learning the target dynamical latent space, conditional variables are extracted from masked neural recordings to perform self-supervised pre-training. To capture temporal evolution in the dynamical latent space of DLFM, conditional features are learned utilizing an autoregressive architecture of transformers. Let the original signal be denoted by $x^p$, and the corresponding masked sample by $\tilde{x}^p$, which shares the same dimensionality as $x^p$. The network for conditional feature extraction is denoted as $F_\alpha$ (with parameters $\alpha$). The conditional variable obtained is written as $c^p = F_\alpha(\tilde{x}^p)$, whose temporal length matches that of $x^p$. In addition, the feature for each time step is denoted as $c(t) \in \mathbb{R}^{k_c}$, where $k_c$ denotes its dimensionality. Conditioned on $c^p(t)$, DLFM employs the flow-based generation to dynamical latent spaces for reconstructing the input masked signal $\tilde{x}^p(t)$. Beyond masked reconstruction, this architecture can be extended to other high-dimensional tasks such as next-token prediction by taking samples from known time steps as input.

**Flow Matching in Dynamical Latent Space** Guided by $c^p(t)$, we establish dynamical latent flow matching for masked reconstruction of $\tilde{x}^p(t)$. Flow matching typically starts from Gaussian noise and iteratively evolves toward the target latent space, following a continuous generation path governed by ordinary differential equations (ODEs) (Lipman et al., 2022). To better capture dynamical

latent patterns, we propose performing flow matching at each time step, focusing on $\tilde{x}^p(t)$ rather than the entire sequence $\tilde{x}^p$. Dynamical evolution information is obtained from $c^p(t)$, providing the foundation for generating the dynamical latent space.

We further denote the dynamical latent variable of flow-based generation conditioned on $c^p(t)$ as $z_t^p(\tau)$. Note that $\tau$ denotes the iteration step along the continuous generative trajectory, rather than the actual time step $t$ of neural spiking activity. For computational convenience, we normalize $\tau$ to the range $[0, 1]$. The flow-based generation of DLFM begins with a Gaussian latent variable $z_t^p(0)$ and evolves towards $z_t^p(1)$, which is trained to approximate the distribution of target dynamical variables $\mathrm{E}_\eta(x^p(t))$. The velocity field of this flow-based generative path along $z_t^p(\tau)$ is parameterized by a neural network $\mathrm{V}_\theta$, where $\theta$ denotes the learnable parameters. The evolution of our dynamical latent variables $z_t^p(\tau)$ over $\tau \in [0, 1]$ is governed by the following ODE:

$$\frac{dz_t^p(\tau)}{d\tau} = \mathrm{V}_\theta\big(z_t^p(\tau), c^p(t), \tau\big). \tag{2}$$

Here, $c^p(t)$ refers to the conditional variables extracted from the input masked sample $\tilde{x}^p$.

During training, $z_t^p(\tau)$ can be explicitly obtained by the linear interpolation between the starting point and the endpoint as $z_t^p(\tau) = (1 - \tau)z_t^p(0) + \tau z_t^p(1)$. Here, $z_t^p(0)$ is initialized with Gaussian noises, and $z_t^p(1) = \mathrm{E}_\eta(x^p(t))$. Moreover, the training of flow matching network $\mathrm{V}_\theta$ aims to approximate the derivative of latent variables along the flow-based generative path, i.e., $z_t(1) - z_t(0)$. We further define the training loss function $\mathcal{L}_{\mathrm{cfm}}(\alpha, \theta)$ for flow-based networks as follows,

$$\mathbb{E}_{t,p(z_t^p(0)),q(z_t^p(1))} \big\| \mathrm{V}_\theta\big(z_t^p(\tau), c^p(t), \tau\big) - \big(z_t^p(1) - z_t^p(0)\big) \big\|^2 . \tag{3}$$

Here, the conditional variable $c^p(t) = \mathrm{F}_\alpha(\tilde{x}^p)[t]$. $p(z_t^p(0))$ and $q(z_t^p(1))$ denote marginal distributions of the initial and final dynamical latent variables along the flow-based generative path. During inference, as the target dynamical latent variable is not directly accessible, dynamical latent variables are inferred by traversing the flow-based generative trajectory conditioned on the extracted features. Specifically, at $\tau = 0$, the dynamical latent variable $z_t^p(0)$ is initialized with Gaussian noises. During the flow-based evolution governed by the ODE in Eq. (2), numerical solvers such as Euler's method or Runge–Kutta schemes are employed to obtain $z_t^p(1)$. This dynamical latent feature then serves as input to either the spiking decoder $\mathrm{R}_\beta$ or the behavioral decoder $\mathrm{G}_\gamma$ for high-dimensional neural decoding.

### 3.2.2 DLFM-BASED PARAMETER-EFFICIENT FINE-TUNING

During the fine-tuning phase, we propose the parameter-efficient fine-tuning utilizing the fixed flow-based generative trajectory. Corresponding to the ODE-based transformation of dynamical latent variables in Eq. (2), the continuous evolution of their conditional distributions can be derived by the Fokker–Planck equation (Lipman et al., 2022). We define the conditional distribution of dynamical latent variables along the pre-trained generative path as $p_\tau(z_t^p(\tau)|c^p(t))$. Its evolution with respect to $\tau$ can be expressed as:

$$\frac{\partial p_\tau(z_t^p(\tau)|c^p(t))}{\partial \tau} = -\nabla \cdot \big(p_\tau(z_t^p(\tau)|c^p(t)) \, \mathrm{V}_\theta(z_t^p(\tau), \mathrm{F}_\alpha(x^p), \tau)\big). \tag{4}$$

Leveraging this probabilistic tractability, the stable transformation path from Gaussian distributions to dynamical latent variables can be established under fixed $\mathrm{V}_\theta$ and conditional variables drawn from similar distributions.

Therefore, we first fine-tune the encoder $\mathrm{E}_\eta$ while keeping the decoders $\mathrm{R}_\beta$ and $\mathrm{G}_\gamma$ fixed on the fine-tuning dataset $\mathcal{D}_f$. This step ensures a consistent mapping to the behavioral space, providing a coarse alignment of the target dynamical latent space. Then, we propose to fine-tune the conditional feature extractor $\mathrm{F}_\alpha$ while keeping $\mathrm{V}_\theta$ fixed to align the conditional variables $c(t)$. Since $\mathrm{V}_\theta$ provides a stable flow-based distribution transformation, aligning $c(t)$ can be approximately interpreted as maximizing the likelihood of the dynamical latent variable distributions on $\mathcal{D}_f$. We thus minimize $\mathcal{L}_{\mathrm{cfm}}(\alpha)$ using $x^f$ to maximize the likelihood, which can be approximately regarded as minimizing the KL divergence between the source and target dynamical latent spaces, as follows:

$$\min_\alpha \mathcal{L}_{\mathrm{cfm}}(\alpha) \approx \max_\alpha \log p_1(z_t^f(1)|c^f(t)) \approx \min_\alpha D_{\mathrm{KL}}\Big(p_1(z_t^f(1)|c^f(t)) \parallel p_1(z_t^p(1)|c^p(t))\Big). \tag{5}$$

This fine-tuning strategy enables stable alignment of dynamical latent spaces under few-shot conditions, owing to parameter-efficient optimization of the mean squared error-based $\mathcal{L}_{\text{cfm}}$, as shown in Table S10. The stable features of dynamical patterns is validated through the visualization analysis presented in Section 4.5. The overall learning procedure of DLFM is presented in Algorithm 1.

---

**Algorithm 1** Dynamical Latent Flow Matching (DLFM)

---

**Input**: pre-training dataset $\mathcal{D}_p$ or fine-tuning dataset $\mathcal{D}_f$; training phase $phase\_m$
**Output**: $F_\alpha$, $V_\theta$, autoencoders $E_\eta$, $R_\beta$, behavioral decoder $G_\gamma$;
 1: Initialize/Load pre-trained $F_\alpha$, $V_\theta$, $E_\eta$, $R_\beta$, $G_\gamma$
 2: **Target Dynamical Latent Space:**
 3: **if** $phase\_m$ is pre-training: **then**
 4:     Update $E_\eta$, $R_\beta$ via Eq. (1); Update $G_\gamma$ using behavioral variables from the dataset;
 5: **else if** $phase\_m$ is fine-tuning: **then**
 6:     Update $E_\eta$ via Eq. (1) and behavioral variables from the dataset;
 7: **end if**
 8: **Conditional Feature Extraction & DLFM**
 9: **for** $iter = 1$ **to** $n_{phase\_m}$ **do**
10:     Sample $x$ from the dataset, obtain $\tilde{x}$ and $c$;
11:     Sample $t$, $\tau$, $z_t(0) \sim \mathcal{N}(0, I)$, $z_t(1) = E_\eta(x(t))$;
12:     **if** $phase\_m$ is pre-training: **then** Update $F_\alpha$, $V_\theta$ via Eq. (3);
13:     **else if** $phase\_m$ is fine-tuning: **then** Update $V_\theta$ via Eq. (3);
14: **end for**
15: **return** $F_\alpha$, $V_\theta$, $E_\eta$, $R_\beta$, $G_\gamma$

---

## 4 EXPERIMENTS AND RESULTS

### 4.1 COMPARATIVE EXPERIMENTS ON SIMULATED DATA

**Experimental Setup** We first conducted experiments on synthetic data to assess the performance of DLFM in few-shot neural adaptation. Following (Kapoor et al., 2024), we employed the Lorenz attractor as the latent dynamics. As illustrated in Fig. 3(a), firing rates were generated via an affine transformation of 3D latent variables into a 96-dimensional space. The 50-time-step synthetic spike trains were then sampled from a Poisson distribution. To further simulate the non-stationarity of spike trains across sessions, we varied the mean firing rates (MFR) or randomly jittered channels while keeping latent dynamics fixed, following the observations reported in (Degenhart et al., 2020) and (Karpowicz et al., 2024).

NDT (Ye & Pandarinath, 2021) and LFADS (Pandarinath et al., 2018) were selected as representative transformer- and VAE-based baselines, respectively. The few-shot adaptation performance was evaluated using bits per spike (bps) Pillow et al. (2008) for high-dimensional spiking prediction. The models were first pre-trained on synthetic data with a fixed MFR of 0.05, followed by few-shot fine-tuning under varying target MFRs and jittered channels. Unless otherwise specified, the default configuration employs 96 channels, 3 shots, and a target MFR of 0.02. The main hyperparameters of DLFM were selected as detailed in Section C.0.3.

**Results** Comparative bps results under varying percentages of jittered channels, shifts in MFRs, and numbers of fine-tuning shots are presented in Fig. 3(b) (Left, Middle, Right), respectively. Each result is averaged over five random runs. We observe that DLFM consistently achieves superior performance under varying degrees of shift, demonstrating its efficiency in few-shot neural adaptation on simulated neural data.

### 4.2 COMPARATIVE EXPERIMENTS ON NLB21

**Experimental Setup** We further evaluated the high-dimensional neural decoding performance of DLFM after self-supervised pre-training on four single-session datasets from NLB21 (Pei et al., 2021), using 5ms bin widths. A brief overview of these datasets is provided in both Section B.1.1 and below:
**dmfc_rsg**: It contains spikes from the dorsomedial frontal cortex of a monkey during ready-set-go

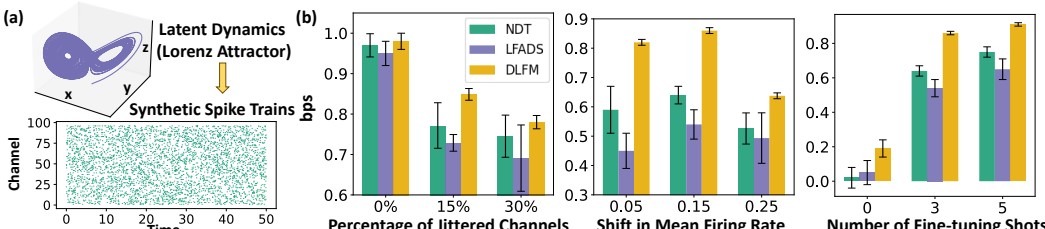

Figure 3: (a) Synthetic spike trains generated from an underlying Lorenz attractor using a Poisson observation model. (b) Comparison of bits-per-spike (bps) performance among NDT, LFADS, and DLFM as a function of percentage of jittered channels (Left), shift in mean firing rate (Middle), and number of fine-tuning shots (Right).

timing tasks.

**area2_bump**: It records spikes from the somatosensory cortex of a monkey during a passive limb perturbation task.

**mc_rtt**: It provides long-duration the primary motor cortex (M1) recordings from a monkey engaged in a random target-reaching task.

**mc_maze**: It offers spikes from M1 as the monkey performs a maze-reaching task.

We compared DLFM with representative baselines for self-supervised representation learning of spiking neural activity, including: (1) STNDT (Le & Shlizerman, 2022), a spatiotemporal transformer model; (2) S5 (Smith et al., 2023), which leverages simplified state-space dynamics; (3) AESMTE3 (Ye & Pandarinath, 2021), a transformer-based ensemble approach; and (4) AutoLFADS (Sedler & Pandarinath, 2023), a variant of LFADS with automated hyperparameter optimization. Evaluation metrics on the latent factors obtained include fp-bps for forward prediction, co-bps for held-out neuron prediction, vel R2 for hand velocity decoding, and psth R2 for trial-averaged neural response reconstruction. Here, we used transformer (Vaswani et al., 2017) as a condition extractor and sit (Peebles & Xie, 2023) as the backbone of flow matching. Self-supervised training was conducted via masked reconstruction, and sampling was based on 3-step Euler integration.

**Results** The best fp-bps results across the four selected datasets, achieved by the baseline methods and our DLFM, are demonstrated in Table 1. The highest result for each dataset is highlighted in bold red, while the second-best is underlined in blue. DLFM demonstrates significantly higher fp-bps in the challenging task of forward time-step prediction.

We further compared the performance of DLFM with more online baselines detailed on the websites listed in Section C.0.1. Our DLFM ranks among the **top two** in fp-bps as shown in Fig. S1. Meanwhile, DLFM exhibits consistent performance on other metrics, such as co-bps and vel R2, as presented in Section C.0.1. These results demonstrate that DLFM achieves superior performance in high-dimensional neural decoding via supervised learning.

Table 1: Best fp-bps (↑) of the baselines and our DLFM on the dmfc_rsg, area2_bump, mc_rtt, and mc_maze datasets with 5 ms bin widths.

|  | STNDT | S5 | AESMTE3 | AutoLFADS | **DLFM** | Total Rank |
|---|---|---|---|---|---|---|
| dmfc_rsg | 0.1910 | 0.1841 | 0.1828 | 0.1960 | **0.1993** | 1/20 |
| area2_bump | 0.1491 | 0.1518 | **0.1603** | 0.1455 | 0.1567 | 2/23 |
| mc_rtt | 0.1260 | 0.1271 | 0.1344 | 0.1240 | **0.1419** | 1/25 |
| mc_maze | **0.2686** | 0.2581 | 0.2589 | 0.2447 | **0.2686** | 1/26 |

## 4.3 COMPARATIVE EXPERIMENTS ON FALCON

**Experimental Setup** To evaluate DLFM in few-shot neural adaptation, we conducted experiments on the M1 and M2 datasets from the Falcon benchmark (Karpowicz et al., 2024). The M1 dataset involves grasp-and-reach tasks accompanied by electromyography (EMG) signals, and the M2 dataset captures fine-grained finger movements with high-resolution kinematic labels. Both datasets comprise multi-session recordings with limited calibration trials, enabling evaluations on few-shot cross-session behavioral decoding.

We compared DLFM with typical baselines employing identical few-shot supervised adaptation (FSS) strategies: Neural Data Transformer 2 (NDT2)(Ye et al., 2023) and NDT3 (Ye et al., 2025). We fine-tuned DLFM using randomly selected trials from each target session, with only 60% and 80% of the original calibration trials, denoted as DLFM-0.6 and DLFM-0.8, respectively. The following results are averaged over 10 random selections. The evaluation metrics for few-shot neural adaptation include EMG or hand kinematic $R^2$ scores on both held-out and held-in sessions (HeldIn$R^2$ and HeldOut$R^2$). Conditional features were also extracted using transformer (Vaswani et al., 2017) combined with session embeddings as contextual input.

Table 2: $R^2$ scores of held-in and held-out sessions on the M1 and M2 datasets. Mean values and standard deviations are reported across sessions.

| Data | Metric | Wiener Filter | NDT2 | NDT3 | **DLFM-0.6** | **DLFM-0.8** | **DLFM** |
|---|---|---|---|---|---|---|---|
| M1 | HeldOut$R^2$ | 0.34 ±0.06 | 0.59 ±0.07 | 0.58 ±0.06 | 0.57 ±0.08 | 0.61 ±0.07 | **0.68 ±0.07** |
| | HeldIn$R^2$ | 0.46 ±0.06 | 0.77 ±0.03 | 0.76 ±0.02 | 0.76 ±0.02 | 0.75 ±0.04 | 0.75 ±0.03 |
| M2 | HeldOut$R^2$ | 0.06 ±0.04 | 0.43 ±0.08 | 0.48 ±0.06 | 0.47 ±0.05 | 0.48 ±0.06 | **0.51 ±0.06** |
| | HeldIn$R^2$ | 0.15 ±0.07 | 0.63 ±0.03 | 0.62 ±0.04 | 0.63 ±0.04 | 0.65 ±0.03 | 0.64 ±0.03 |

**Results** The main results on HeldOut$R^2$ and HeldIn$R^2$ averaged over both sessions and runs are presented in Table 2. The highest result for each dataset is highlighted in bold red, while the second-best is underlined in blue. We find that DLFM attains substantially higher HeldOut$R^2$ on both datasets even with only 80% of the calibration trials, and achieves competitive performance even with 60%. The increase in HeldOut$R^2$ does not come at the expense of decreased HeldIn$R^2$, indicating the generalization capability of the learned dynamical latent space. Therefore, the proposed DLFM demonstrates superiority in efficient few-shot neural adaptation.

### 4.4 Ablation Experiments on Main Components

We conducted ablation studies on the main components of both pre-training and fine-tuning phases, as shown in Fig. 4(a). During the pre-training phase, we compared DLFM with three variants: DLFM-W, which performs flow matching directly on full input signal sequences; DLFM-R, which replaces the conditional feature extractor with a recurrent neural network (RNN); and DLFM-M, which employs a multilayer perceptron (MLP) as the flow-based backbone. For the fine-tuning phase, we considered two variants: DLFM-F, which fine-tunes both $F_\alpha$ and $V_\theta$, and DLFM-V, which fine-tunes $V_\theta$ while keeping $F_\alpha$ fixed. The dmfc_rsg and mc_rtt datasets were selected as representative datasets of passive and active motor tasks, respectively.

As shown in Fig. 4(b), decoding performance on the dmfc_rsg and mc_rtt datasets substantially decreases with the full sequence input, indicating that flow-based models may not be well-suited for directly learning dynamical patterns. The pronounced performance drop of DLFM-R highlights the importance of efficiently incorporating temporal evolution into the conditional variables, while the comparative performance of DLFM-M illustrates the flexibility in choosing flow-based networks. HeldOut$R^2$ of DLFM-F shows a slight drop, while DLFM-V decreases more on few-shot adaptation of the M1 and M2 datasets. This demonstrates the effectiveness of our parameter-efficient fine-tuning, as shown in Fig. 4(c).

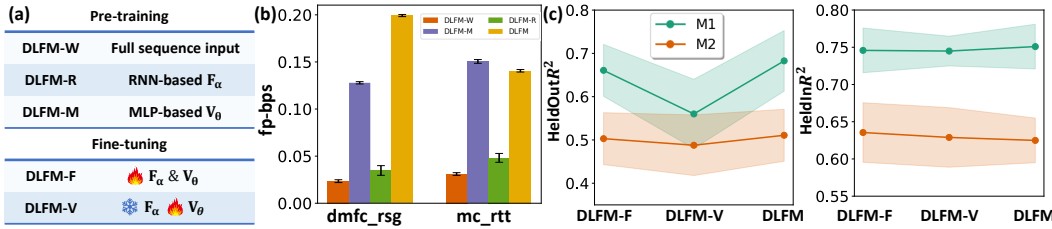

Figure 4: (a) Overview of DLFM variants across the pre-training and fine-tuning phases. (b) Ablation performance of DLFM-W, DLFM-R, DLFM-M, and DLFM on the dmfc_rsg and mc_rtt 5ms datasets. (c) Few-shot adaptation performance, with HeldOut$R^2$ (Left) and HeldIn$R^2$ (Right) for DLFM-F, DLFM-V, and DLFM on the M1 and M2 datasets.

## 4.5 Interpretability Analysis of DLFM

To examine the dynamical latent space for neural adaptation, we performed a interpretability analysis of the dynamical latent variables by DLFM on both the synthetic datasets and Falcon. Since the Lorenz attractor (Lorenz, 1963) is widely used to model neural dynamics (Pandarinath et al., 2018; Brenner et al., 2024), we used the aforementioned Lorenz-based synthetic data. We employed $R^2$ scores to evaluate the latent dynamics captured by obtained latent variables. As shown in Fig. 5(a) and (b), as well as in Fig. S2, the DLFM exhibits more consistent and accurate latent dynamical trajectories under few-shot neural adaptation. In addition, we computed the mutual information to quantify the consistency between dynamical latent variables extracted from HeldIn and HeldOut sessions, as shown in Fig. 5(c). We further visualized the latent variables of M1 using PCA, as shown in Fig. 5(d), and those of M2 in Fig. S3. These results indicate that DLFM effectively captures stable features of the underlying dynamical patterns, therefore enhancing few-shot neural adaptation. Meanwhile, the superiority of DLFM in neural adaptation does not come at the expense of computational time (pre-training, fine-tuning, and inference), as shown in Section C.0.4.

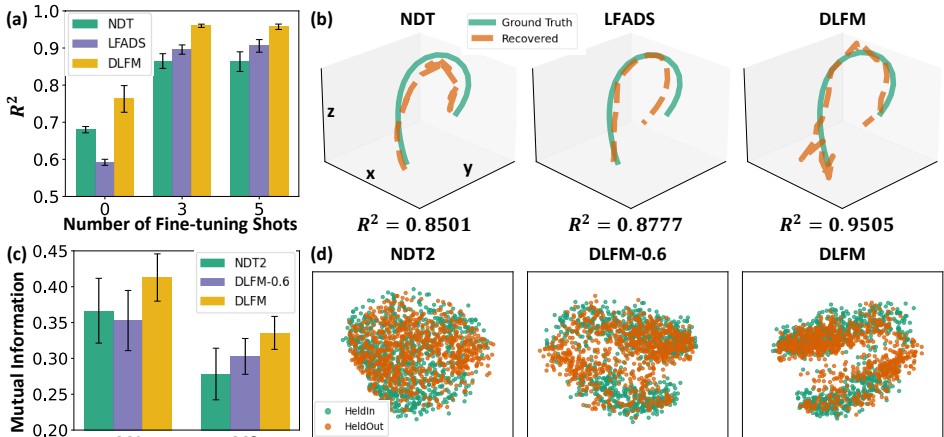

Figure 5: (a) $R^2$ scores of latent dynamical recovery by NDT, LFADS, and DLFM with different fine-tuning shots. (b) Recovered latent dynamics and $R^2$ scores by NDT, LFADS, and DLFM. The target mean firing rates are set to 0.1 under the 3-shot setting. Green lines denote the ground-truth latent trajectories, while orange dashed lines indicate the recovered dynamics. (c) Mutual information between held-in and held-out features by NDT2, DLFM-0.6, and DLFM for few-shot adaptation. (d) The PCA visualization of latent features from NDT2, DLFM-0.6 and DLFM on M1. The visualized features are randomly selected from held-in (green) and held-out (orange) sessions.

## 5 Conclusion and Limitations

In this study, we introduce a novel neural decoding framework of Dynamical Latent Flow Matching (DLFM) designed for efficient few-shot neural adaptation. Our DLFM performs flow matching in the dynamical latent space through probabilistic flexibility and tractability, achieving efficient few-shot adaptation. Further interpretability analysis on both the model of Lorenz attractor and the Falcon benchmark demonstrates that DLFM successfully captures intrinsic features of dynamical patterns within the neural manifold. Our framework achieves superior decoding performance on both simulated data and the NLB21 benchmark, ranking among the top two in forward prediction tasks across all web submissions. Additionally, DLFM exhibits high-performance over few-shot adaptation on the Falcon, achieving competitive performance using only 60% of the calibration trials. These findings underscore the potential of DLFM as a reliable and high-performance neural decoding framework, advancing the practical deployment of BCI systems.

**Limitations** Several limitations of this work deserve further investigation. The effectiveness of DLFM in more complex scenarios, such as multi-modal datasets from distributed brain regions, remains to be explored. In addition, the application of DLFM to zero-shot or fully unsupervised adaptation remains an open and promising direction for future investigation.

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

## A   METHODOLOGY

### A.1   DETAILED ARCHITECTURE OF DLFM

We present the detailed architecture of our main modules as follows. The input neural signals have the shape of (Batch size=256, Signal temporal length=$m$, Number of channels=$l$). The latent dimensions of conditional features $c(t)$ are denoted as $k_c$, the dimension of latent states in the spatially coupled flow matching is the same, $k_c$. The depth of the transformer layers in $F_\alpha$ is denoted as $n_{cd}$, and the depth of the transformer layers in $V_\theta$ is denoted as $n_{fd}$. The dropout value is represented as $o_d$. The architectures of $F_\alpha$, and $V_\theta$ can be seen in Table S1.

Table S1: Detailed Architectures of DLFM

| | |
|---|---|
| $F_\alpha$ | $[\text{MSA}(k_c, n_{head}), \text{FFN}(k_c \times n_{head}, k_c)] \times n_{cd}$ |
| $V_\theta$ | $[\text{FFN}(k_c \times 2, k_c), \text{MSA}(k_c, n_{head}), \text{MLP}(k_c \times n_{head}, k_c)] \times n_{fd}$ |

Here, we use the term MLP to refer to Multilayer perceptron with residual connections, MSA to represent multi-head self-attention modules, and FFN to indicate feed-forward neural networks.

Moreover, default dimensions $k_c$, the drop-out rate $v_d$, the number of heads $n_{head}$ and the network depth $n_{cd}$ and $n_{fd}$ mentioned above are configured as shown in Table S2 according to different experimental datasets.

Table S2: Default Value Setup on Different Experimental Datasets

| | $k_c$ | $n_{fd}$ | $n_{cd}$ | $v_d$ | $n_{head}$ |
|---|---|---|---|---|---|
| Simulated Data | 96 | 5 | 2 | 0.1 | 2 |
| NLB21 | 64 | 5 | 2 | 0.1 | 2 |
| Falcon | 32 | 5 | 2 | 0.1 | 8 |

The default values for the simulated data are set according to the results reported in Table S9. For the other datasets, $k_c$ is selected based on the configurations used in (Ye & Pandarinath, 2021) and (Ye et al., 2023), given the similarity of the datasets. The remaining hyperparameters are determined through grid search, and were found to have minimal influence on the overall performance of DLFM.

### A.2   ADDITIONAL RELATED WORK ON FLOW MATCHING

Normalizing flows have been widely applied in distribution sampling due to their precise and explicit likelihood modeling. Traditional normalizing flows (Chen et al., 2019; Dinh et al., 2022) typically rely on invertible transformations, but these can constrain the representational capacity of the networks. Recent research has sought to alleviate this limitation by utilizing continuous normalizing flows (Yang et al., 2019) based on ODEs. For example, flow matching (Lipman et al., 2022; Peebles & Xie, 2023) extends diffusion models, an advanced generative model, allowing for more flexible diffusion paths. Conditional flow matching (Atanackovic et al., 2025) further incorporates conditional features to model conditional distributions.

# B  EXPERIMENTAL DETAILS

## B.1  DATASET DESCRIPTION

### B.1.1  NLB 21

The Neural Latents Benchmark 2021 (NLB 21) (Pei et al., 2021) is a comprehensive benchmark suite designed to evaluate latent variable models on neural population spiking data. The central task involves unsupervised co-smoothing, where models predict the firing rates of held-out neurons based on observed neural activity, enabling systematic comparison across diverse brain areas and behavioral paradigms. NLB 21 comprises four datasets collected from non-human primates performing motor, sensory and cognitive tasks, which have been introduced in the main text.

All datasets are provided in the Neurodata Without Borders (NWB) standard format and are accessible via the DANDI archive. The raw spike trains are typically binned into 5ms non-overlapping windows to convert discrete spike events into firing rate estimates suitable for latent variable modeling. Trials with insufficient duration or excessive noise are excluded based on dataset-specific criteria defined in the benchmark documentation. Behavioral and auxiliary signals, when available (e.g., kinematics, force), are synchronized with neural data and preprocessed by standard normalization techniques to ensure compatibility with modeling pipelines.

The benchmark uses the bits per spike (bps) metric within an unsupervised co-smoothing framework as the primary evaluation criterion. This metric quantifies how accurately models reconstruct the activity of held-out neurons from observed population activity. Submissions are evaluated on the EvalAI platform, which provides a leaderboard and enforces standardized scoring procedures. An official Python toolkit, `nlb_tools`, facilitates data loading, preprocessing, model evaluation, and includes baseline implementations to support reproducibility and rapid experimentation. This benchmark suite offers a rigorous platform to assess the ability of latent variables to capture complex neural population dynamics across multiple brain regions, behavioral contexts, and data regimes.

### B.1.2  FALCON

The Falcon dataset (Karpowicz et al., 2024) is a large-scale, high-quality neural recording dataset designed to facilitate the study of neural population dynamics during naturalistic, free-behavior tasks. Recorded from non-human primates, Falcon provides extensive spike train data collected while subjects engage in complex, unconstrained movements.

Data are provided in standardized formats compatible with common neuroscience analysis toolkits. Neural spike trains are binned into non-overlapping windows to convert discrete spikes into continuous firing rate estimates suitable for latent variable modeling. Behavioral signals are preprocessed with standard normalization and temporal alignment procedures. Falcon offers a challenging testbed for latent variable models aiming to capture high-dimensional neural population activity in naturalistic conditions. The dataset enables evaluation of model robustness beyond traditional trial-based paradigms, facilitating progress towards understanding neural coding in real-world contexts. The M1 and M2 dataset used in our paper are available from `https://dandiarchive.org/dandiset/000941?search=falcon&pos=1` and `https://dandiarchive.org/dandiset/000953?search=falcon&pos=2`, respectively.

## B.2 TRAINING DETIALS

The main configurations for model training included the learning rate, weight decay parameters of the Adam optimizer, batch sizes, number of iterative epochs, and random seeds during pre-training and fine-tuning phases. The epochs of pre-training and fine-tuning use default settings with early stopping. The best-trained weights are selected based on performance on the validation set, which typically comprises 20% of the training data. Details of these hyperparameters are provided in Table S3 and Table S4, respectively.

Table S3: Detailed Pre-training Setup

|                | Learning Rate | Weight Decay | Epochs | Batch Size | Random Seed |
|----------------|---------------|--------------|--------|------------|-------------|
| Simulated Data | 2e-3          | 1e-5         | 400    | 256        | 0-4         |
| NLB21          | 2e-3          | 1e-5         | 1200   | 256        | 0-4         |
| Falcon         | 2e-3          | 1e-5         | 2000   | 256        | 0-4         |

Table S4: Detailed Fine-tuning Setup

|                | Learning Rate | Weight Decay | Epochs | Batch Size | Random Seed |
|----------------|---------------|--------------|--------|------------|-------------|
| Simulated Data | 2e-3          | 1e-5         | 80     | 256        | 0-4         |
| Falcon         | 2e-3          | 1e-5         | 1000   | 256        | 0-4         |

In addition, the pre-training and fine-tuning of SCFM on the simulated data and NLB21 datasets are conducted using an NVIDIA GeForce RTX 3080 Ti (12GB). For the Falcon dataset, pre-training is performed on four NVIDIA A800 GPUs (80GB each), and fine-tuning is carried out on a single A800.

## B.3 VALIDATION DETAILS

Specifically, during the validation after fine-tuning phases, we employed neural signals from the test datasets, which were not leveraged during the fine-tuning phase, to evaluate the efficacy of our DLFM.

The evaluation is conducted based on the decoding performance from the generated latent variables. Specifically, we first sample $z_t(1)$ using a 3-step Euler method starting from $z_t(0)$, guided by the velocity field $V_\theta(z_t(0), 0, F_\alpha(\tilde{x}))$. The predicted target label $y(t)$ is then computed as $y(t) = G_\gamma(z_t(1))$. The reconstructed or predicted neural recordings are further obtained as $x(t) = R_\beta(z_t(1))$.

# C   ADDITIONAL RESULTS

## C.0.1   ADDITIONAL COMPARATIVE RESULTS ON NLB21

The web submissions for NLB21 (Pei et al., 2021) are available on the EvalAI platform. The leaderboards summarizing all submissions for each dataset are presented below:
`https://eval.ai/web/challenges/challenge-page/1256/leaderboard/`
`3185`: **dmfc_rsg**
`https://eval.ai/web/challenges/challenge-page/1256/leaderboard/`
`3186`: **area2_bump**
`https://eval.ai/web/challenges/challenge-page/1256/leaderboard/`
`3187`: **mc_rtt**
`https://eval.ai/web/challenges/challenge-page/1256/leaderboard/`
`3188`: **mc_maze**
A snapshot of the leaderboard for fp-bps across the four nlb datasets is shown in Fig. S1.

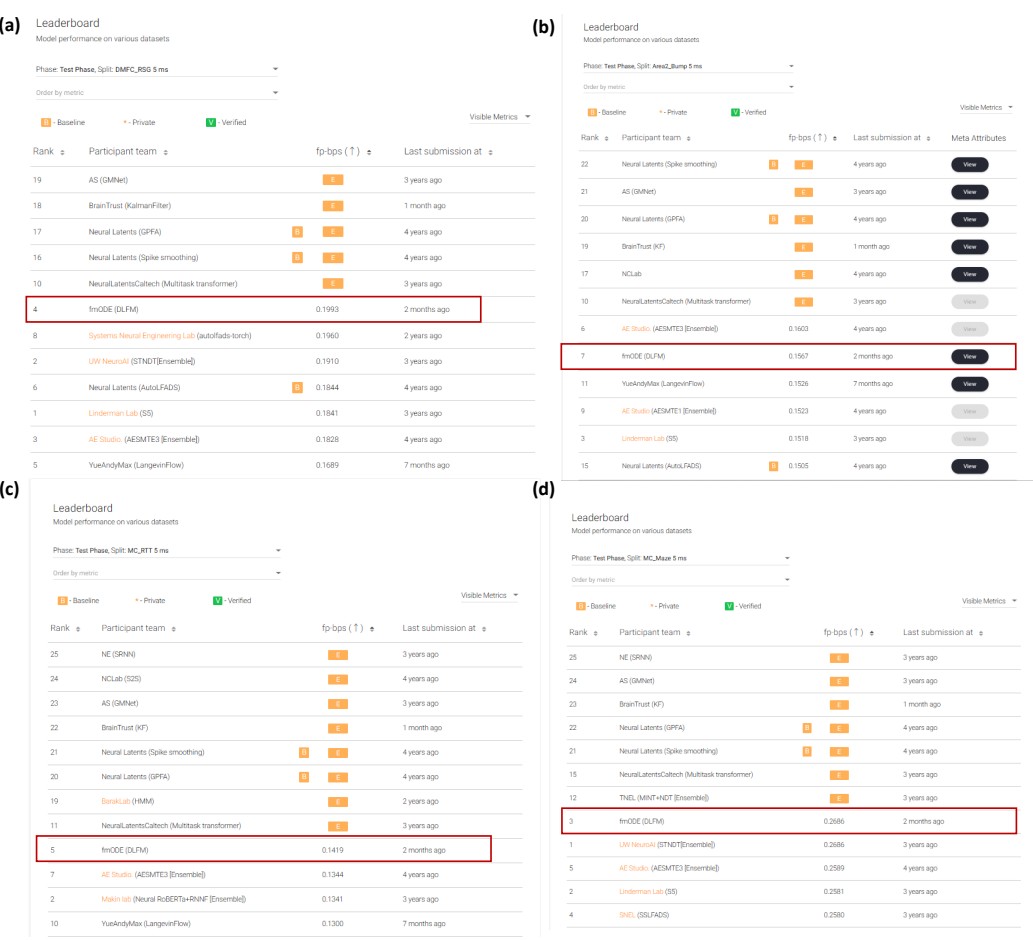

Figure S1: Snapshot of the fp-bps leaderboard for (a) dmfc_rsg, (b) area2_bump, (c) mc_rtt, and (d) mc_maze datasets. Our method is reported as fmODE (DLFM).

The comprehensive results on various metrics are shown in Table S5 (dmfc_rsg), Table S6 (area2_bump), Table S7 (mc_rtt) and Table S8 (mc_maze), correspondingly. Colored blocks indicate the top three values in each column, with darker shades representing better values.

Table S5: Comprehensive comparison of DLFM with baselines on dmfc_rsg dataset.

| | co-bps (↑) | psth R2 (↑) | fp-bps (↑) |
|---|---|---|---|
| STNDT | 0.1940 | 0.6452 | 0.1910 |
| S5 | 0.2020 | 0.4407 | 0.1841 |
| AESMTE3 | 0.1886 | 0.6064 | 0.1828 |
| AutoLFADS | 0.1820 | 0.5873 | 0.1960 |
| **DLFM** | 0.1869 | 0.6373 | 0.1993 |

Table S6: Comprehensive comparison of DLFM with baselines on area2_bump dataset.

| | co-bps (↑) | vel R2 (↑) | psth R2 (↑) | fp-bps (↑) |
|---|---|---|---|---|
| STNDT | 0.2904 | 0.8937 | 0.7303 | 0.1491 |
| S5 | 0.2901 | 0.8727 | 0.7258 | 0.1518 |
| AESMTE3 | 0.2860 | 0.8999 | 0.7109 | 0.1603 |
| AutoLFADS | 0.2535 | 0.8516 | 0.6104 | 0.1455 |
| **DLFM** | 0.2844 | 0.8820 | 0.7199 | 0.1567 |

Table S7: Comprehensive comparison of DLFM with baselines on mc_rtt dataset.

| | co-bps (↑) | vel R2 (↑) | fp-bps (↑) |
|---|---|---|---|
| STNDT | 0.2065 | 0.6352 | 0.1260 |
| S5 | 0.2296 | 0.6450 | 0.1271 |
| AESMTE3 | 0.2053 | 0.6334 | 0.1344 |
| AutoLFADS | 0.1882 | 0.6176 | 0.1240 |
| **DLFM** | 0.2073 | 0.6663 | 0.1419 |

Table S8: Comprehensive comparison of DLFM with baselines on mc_maze dataset.

| | co-bps (↑) | vel R2 (↑) | psth R2 (↑) | fp-bps (↑) |
|---|---|---|---|---|
| STNDT | 0.3862 | 0.9095 | 0.6693 | 0.2686 |
| S5 | 0.3823 | 0.9043 | 0.6431 | 0.2581 |
| AESMTE3 | 0.3676 | 0.9114 | 0.6683 | 0.2589 |
| AutoLFADS | 0.3497 | 0.9027 | 0.6170 | 0.2447 |
| **DLFM** | 0.3779 | 0.9019 | 0.6525 | 0.2686 |

### C.0.2 ADDITIONAL RESULTS ON INTERPRETABILITY ANALYSIS

As shown in Fig. S2, we visualized the recovered trajectories along with their corresponding $R^2$ under the 5-shot adaptation. Compared to NDT and LFADS, DLFM produces more consistent and accurate reconstructions, closely matching those achieved under 3-shot adaptation. This underscores its effectiveness in capturing stable features of dynamical patterns and enabling efficient few-shot neural adaptation.

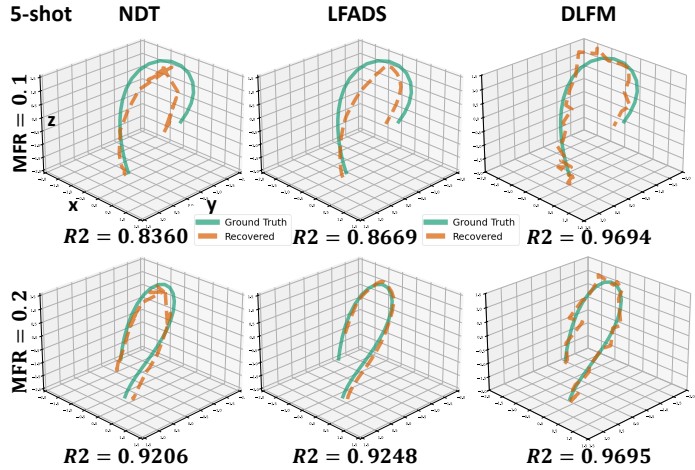

Figure S2: Recovered latent dynamics and $R^2$ scores achieved by NDT, LFADS, and DLFM on target datasets. The target mean firing rates (MFR) are set to 0.1 and 0.2 under the 5-shot setting, respectively. Green lines denote the ground-truth latent trajectories, while orange dashed lines indicate the recovered dynamics.

We further visualized the latent features obtained by DLFM-0.4, DLFM-0.6, DLFM-0.8 and DLFM using randomly selected trials from held-in and held-out sessions of the M2 dataset. These features were projected into two dimensions via Principal Component Analysis (PCA). As shown in Fig. S3, DLFM exhibits structured latent distributions and alignment between held-in and held-out sessions even with fewer calibration trials. This indicates that DLFM preserves stability within the latent space, thereby enabling consistent neural decoding in few-shot scenarios.

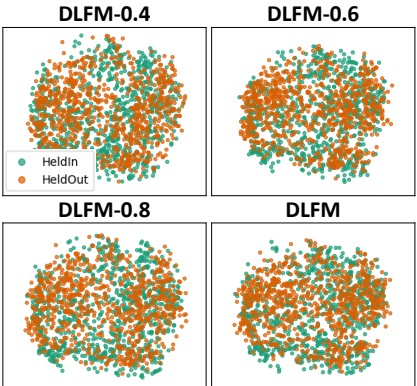

Figure S3: The PCA visualization of latent features generated by DLFM-0.4, DLFM-0.6, DLFM-0.8 and DLFM on M2. The visualized features are randomly selected from held-in (green) and held-out (orange) sessions. DLFM-0.4, DLFM-0.6, and DLFM-0.8 denote partial fine-tuning using 40%, 60%, and 80% of the original calibration trials, respectively.

### C.0.3 HYPER-PARAMETER SENSITIVITY ANALYSIS

The main hyperparameters of our DLFM method include the latent variable dimension $k_c$, the depth of the transformer layers in the flow matching module $n_{fd}$, and the depth of the transformer-based conditional feature extractor $n_{cd}$. In this experiment, we analyzed the performance of DLFM on the synthetic datasets under different configurations of these main hyperparameters. This grid-search was performed by varying each parameter individually while keeping the others fixed at their default values as mentioned in Table S2. As shown in Table S9, excessively high latent dimensionality can introduce representational redundancy and lead to a decline in bits-per-spike (bps) performance. Additionally, variations in the number of transformer layers, denoted as $n_{fd}$ and $n_{cd}$, have minimal impact on bps results. Based on these observations, we adopt default values of $k_c = 96$, $n_{fd} = 5$, and $n_{cd} = 2$ for our experiments.

Table S9: Bits-per-spike (bps) performance under varying hyperparameters: $k_c$, $n_{fd}$, and $n_{cd}$ on simulated source neural data with a mean firing rate of 0.05. The reported mean and standard deviation are computed over five independent runs.

| $k_c$ | 64 | 96 | 128 |
|---|---|---|---|
| bps | $0.95 \pm 0.01$ | $\mathbf{0.96} \pm 0.02$ | $0.90 \pm 0.05$ |
| $n_{fd}$ | 4 | 5 | 6 |
| bps | $0.95 \pm 0.02$ | $\mathbf{0.96} \pm 0.02$ | $0.95 \pm 0.02$ |
| $n_{cd}$ | 1 | 2 | 3 |
| bps | $0.97 \pm 0.02$ | $\mathbf{0.96} \pm 0.02$ | $0.94 \pm 0.01$ |

### C.0.4 COMPUTATIONAL ANALYSIS

The computational analysis of DLFM is presented in terms of runtime and parameter count. The number of parameters in the core components, the conditional feature extractor $F_\alpha$ and the flow-based model $V_\theta$, is summarized in Table S10. As expected, the total number of parameters increases with dataset complexity and scale. Notably, $F_\alpha$ constitutes only a subset of the total model parameters, supporting our claim that fine-tuning can be efficiently performed on small subsets of DLFM.

Table S10: The number of parameters in the conditional feature extractor $F_\alpha$, the flow-based model $V_\theta$, and the total parameter count of DLFM on the simulated neural data, NLB21, and Falcon datasets.

| Data | $F_\alpha$ | $V_\theta$ | Total |
|---|---|---|---|
| Simulated Data | 0.1M | 0.3M | 0.5M |
| NLB21 | 0.8M | 3.1M | 4.5M |
| Falcon | 12.6M | 5.1M | 32.9M |

As shown in Table S11, DLFM achieves rapid fine-tuning with few-shot data, requiring less than 20 minutes (min) on the multi-session Falcon dataset. Furthermore, the inference time per sample is under 20 milliseconds (ms), highlighting the potential of the DLFM framework for real-time or online neural decoding applications.

Table S11: Computational time (minutes (min)/milliseconds (ms)) of DLFM during pre-training, fine-tuning and inference on simulated neural data, NLB21 and Falcon datasets.

| Data | Pre-training (min) | Fine-tuning (min) | Inference (ms) | Device |
|---|---|---|---|---|
| Simulated Data | 15 | 0.2 | 4 | 3080Ti |
| NLB21 | 82 | – | 11 | 3080Ti |
| Falcon | 571 | 17 | 18 | A800 |

