# OpenReview forum: "Dynamical Latent Flow Matching for High-Performance Few-Shot Neural Adaptation"
_ICLR.cc/2026/Conference — ICLR 2026 Conference Withdrawn Submission_

### Official Review · Reviewer_zBHA · 2025-10-20

**Soundness:** 2
**Presentation:** 2
**Contribution:** 2
**Rating:** 2
**Confidence:** 5

**Summary:**

The authors suggest a new method to learn neural dynamics by applying flow matching in latent space to learn dynamics. Through real world benchmark neural recording data and synthetic data, the author demonstrates effective latent dynamics learning by the proposed method. By freezing the flow matching module that handles stochastic, unstable components in dynamics and fine-tuning encoder and feature extraction parts, the authors empirically show that such fine-tuning can focus on stable components of dynamics and thus improve data efficiency in calibrating new sessions. However, the suggested work shows high similarity of the previously published work and several caveats elaborated in Weaknesses and Questions should be resolved.

**Strengths:**

* Successful application of well accepted flow matching to latent dynamic modeling of neural recordings.

* Demonstration of effective forward prediction in several datasets learned by the proposed models.

* Interesting few shot adaptation strategy which is empirically validated.

**Weaknesses:**

* The flow matching idea is already applied and tested for latent neural dynamics and few shot adaptation by Wang et al., 2025 [1], with a very similar approach and goal (titled "Flow Matching for Few-Trial Neural Adaptation with Stable Latent Dynamics"). The novelty over this prior work should be sufficiently suggested, and the difference should be thoroughly analyzed.

* The evaluation scheme seems incomplete. The only forward prediction is tested for Neural Latent Benchmark while the benchmark includes official behavioral decoding tasks. For Falcon benchmark, it is unclear why H1, H2, and B1 from the original dataset are not tested.

* The results are mixed with other metrics. As shown in Table S5-8, the baselines are often better in other evaluation metrics. The “fp-bps” that shows the best results for the proposed model is only presented in the main paper and other inferior results are in the supplementary. The decision for this should be sufficiently explained to provide a fair viewpoint, otherwise all metrics should be presented upfront equally.
It will be better to provide more rationales on why stable dynamics can be learned by flow matching.  Currently, more of technical description of definition of the ideas is majorly described, but the main key idea that how the flow matching could contribute to learning stable dynamics is not well explained. As this claim is stated multiple times, having more detailed logical explanations will benefit the readers.

* The paper lacks details on the baseline model implementation, especially on Falcon benchmark in Table 2. The reason for asking is that the numbers reported in the original Falcon paper (Table 1 in Karpowicz et al.) for the baseline model, NDT2, are different from those reported by the authors. More details are needed to fairly evaluate the results.

[1] Wang et al., Flow Matching for Few-Trial Neural Adaptation with Stable Latent Dynamics. ICML 2025

**Questions:**

* While parameter efficiency is claimed, the number or proportion of the parameters that are fine-tuned is not well described.

* In section 3.2.1, the target dynamical latent space is obtained following Rombach et al., 2022. I was wondering how the authors are following the original paper that used VAE (either quantized or not) for autoencoders. Moreover, this was for an image which doesn’t have any temporal dynamics in it. It is not well explained how the dynamical latent is achieved.

* In Section 4.5 and Figure 5, I was wondering how the model comparison looks like if the person correlation is used since qualitatively, LFADS is producing more matching dynamics in Fig 5 (b).

* In Table 2, how does the model behave with 10%, 20%, and 40% of the calibration data? Also, how do the baselines perform along those ratios from 10% to 100%?

* The using 60% in M1 shows severe drop in performance, thus hard to say it is competitive. More explanation is needed on this claim.

---

### Official Review · Reviewer_ysBq · 2025-10-31

**Soundness:** 2
**Presentation:** 2
**Contribution:** 3
**Rating:** 4
**Confidence:** 3

**Summary:**

The authors designed a new model called Dynamical Latent Flow Matching (DLFM) for consistent, long-term motor decoding. Their model operates on latent neural dynamics using a flow-matching method. They benchmarked their model against other models on two different evaluation datasets.

**Strengths:**

The study investigates an important question in the motor-decoding field and will have great value for applications such as BCIs.

Using flow matching in latent space is novel for motor decoding.

DLFM achieves top-two performance in the NLB-2021 benchmark.

DLFM achieves the best performance in the FALCON benchmark.

**Weaknesses:**

The improvements of DLFM over other models in the NLB-21 benchmark are very small (Table 1) and only for one metric (fp-bps, Table S5-S8). In contrast, the performance gain in the FALCON benchmark is huge (Table 2). The authors have submitted their NLB-21 results to the EvalAI platform (Figure S1). Unfortunately, they haven’t submitted their FALCON results to the same EvalAI benchmark. I will increase my score once these results are uploaded to the platform and verified.

**Questions:**

Do the authors use all the training data to train their model? I would like to know how their model performs with different numbers of training days, similar to Figure 3B of the SPINT paper (Le et al., 2025).

---

### Official Review · Reviewer_mhBV · 2025-10-31

**Soundness:** 2
**Presentation:** 3
**Contribution:** 2
**Rating:** 2
**Confidence:** 4

**Summary:**

The authors present a method for few-shot model adaptation applied to neural data using diffusion methods. They use flow-matching in a neural latent space. They evaluate on common neural data benchmarks for BCI decoding and in simple simulations.

**Strengths:**

- The overall framework is outlined nicely (good Fig 2 diagram, clear equations).
- Datasets are standard benchmarks for motor BCI tasks (Falcon, NLB) and a fair number of model comparisons were implemented.
- Ablation studies done are good.

**Weaknesses:**

- Description of the method leaves it unclear as to which parts are novel and which are established in the literature.
- Comparison methods are a bit out of date (LFADS, NDT) compared to other possibilities (POYO, NDT2 or NDT3).
- Figure 1 and the problems in current models are not clearly described. What is negative transfer in Fig 1b? And training instability in Fig 1d? If these are problems the paper intends to solve, I would expect later figures to also use these metrics and prove the new approach improves them.
- Number of fine-tuning trials axis being flipped compared to fine-tuning epochs is a bit hard to read.
- Table 1 lacks any variance estimation or statistical comparison between models.
- Table 2 has these but do not correctly identify when there is a statistically significant difference.
- Figure 4 in ablation studies do not seem to have significant differences in R2 but show strong differences in bps; aka no diff in fine tuning but strong diff in model variant. It is confusing to have different metrics evaluated for different models.

**Questions:**

- How does this method compare to Wang, P., Qi, Y., Wang, Y., & Pan, G. Flow Matching for Few-Trial Neural Adaptation with Stable Latent Dynamics. (ICML, 2025)?
- Why was "few-shot adaptation performance ... evaluated using bits per spike (bps) Pillow et al. (2008) for high-dimensional spiking prediction."? Why was R2 used in other cases? What can be learned about the model from these different metrics?
- 5 ms bin widths were chosen. Is this standard? Other works use much longer bins (20-30 ms).
- How are intrinsic features captured/quantified? R2 of lorenz dynamics isn't generalization to real data.
- Does Fig 5c show significant differences? What is the interpretation of Fig 5d? Is Figure S2 supposed to be the Falcon dataset?

**Details Of Ethics Concerns:**

This seems somewhat similar to an already published paper accepted in ICML 2025: https://openreview.net/forum?id=nKJEAQ6JCY
It does cover new content but it's concerning to have text replicated (abstract, intro, ...).

For example, the first sentences are "The goal of brain-computer interfaces (BCIs) is to establish a direct connection between the brain
and external devices, offering promising avenues for neural rehabilitation in individuals with paralysis (Willett et al., 2021; Metzger et al., 2023; Willett et al., 2023)." (this work) and "The aim of Brain-computer Interfaces (BCIs) is to establish a direct link between the brain and external devices, presenting great opportunities for improving neural rehabilitation in individuals with paralysis (Willett et al., 2021; 2023;
Wu et al., 2016; Wang et al., 2023a)." (Wang, 2025).

Similar figure styles (Fig 2) indicate perhaps coming from the same lab, which could explain some of the possible-plagiarism phrasing.

---

### Official Review · Reviewer_nzks · 2025-11-01

**Soundness:** 3
**Presentation:** 3
**Contribution:** 2
**Rating:** 8
**Confidence:** 3

**Summary:**

In this work, the authors proposed a framework named the Dynamical Latent Flow Matching (DLFM), which works for addressing the challenges of few-shot neural adaptation in Brain-Computer Interfaces (BCIs) due to the drastic distribution shift recorded across sessions. Existing standard neural latent variable modeling methods like LFADS and Transformer-based MtM often suffer from performance degradation as they were not naturally built to maintain the robustness across sessions. DLFM relieves this  issue by performing conditional diffusio nmodel flow matching in the dynamical neural latent space, which leverages the stable dynamical patterns preserved within the neural manifold across the pre-training sessions and the fine-tuning sessions.

In the experimental section, the effectiveness of the proposed DLFM is evaluated on the wide-adopted NLB'21 benchmark with interpretable analysis.

**Strengths:**

1. The overall paper is well-written and in a concrete flow, the figures in the paper are vivid and illustrates the framework clearly.
2. I think the use of diffusion flow-matching method here is a good contribution for the few-shot learning task in across-session neural data learning. As mentioned in the paper, diffusion models get rid of the unflexible VAE priors and also provides density estimation power through the SDE/ODE flow.
3. The experimental results in the paper are solid across simulated Lorenz data and also the NLB'21 benchmark. The baseline comparisons are abundant.

**Weaknesses:**

1. Flow-matching for modeling the dynamical latent dynamics itself seems not a too novel technique at this moment, the reason of choosing it and comparisons of it with some of its variants should be further clarified.
2. A small arrangement notice of the text would suggest the algorithm 1 to be put in the appendix while putting more experimental results to the main text.

**Questions:**

1. Could you tell us what's the key intuitive/motivation that the diffusion flow-matching would work well on the few-shot learning/domain adaptation tasks?

I have no other questions, other concerns please refer to my Weaknesses section.

---

### Note · Authors · 2026-01-16

I have read and agree with the venue's withdrawal policy on behalf of myself and my co-authors.